# Occurrence and Temporal Variability of Out-of-Hospital Cardiac Arrest during COVID-19 Pandemic in Comparison to the Pre-Pandemic Period in Poland—Observational Analysis of OSCAR-POL Registry

**DOI:** 10.3390/jcm11144143

**Published:** 2022-07-16

**Authors:** Jakub Ratajczak, Stanisław Szczerbiński, Aldona Kubica

**Affiliations:** 1Department of Health Promotion, Nicolaus Copernicus University, Collegium Medicum in Bydgoszcz, 85-094 Bydgoszcz, Poland; aldona.kubica@gmail.com; 2Department of Cardiology and Internal Medicine, Nicolaus Copernicus University, Collegium Medicum in Bydgoszcz, 85-094 Bydgoszcz, Poland; 3Emergency Medical Center in Opole, 45-369 Opole, Poland; pielegniarz@tlen.pl

**Keywords:** temporal variability, COVID-19, coronavirus, out-of-hospital cardiac arrest (OHCA), sudden cardiac death

## Abstract

An investigation of the chronobiology of out-of-hospital cardiac arrest (OHCA) during the coronavirus disease 2019 (COVID-19) pandemic and the differences in comparison to the 6-year pre-pandemic period. A retrospective analysis of the dispatch cards from the Emergency Medical Service between January 2014 and December 2020 was performed within the OSCAR-POL registry. The circadian, weekly, monthly, and seasonal variabilities of OHCA were investigated. A comparison of OHCA occurrence between the year 2020 and the 6-year pre-pandemic period was made. A total of 416 OHCAs were reported in 2020 and the median of OHCAs during the pre-pandemic period was 379 (interquartile range 337–407) cases per year. Nighttime was associated with a decreased number of OHCAs (16.6%) in comparison to afternoon (31.5%, *p* < 0.001) and morning (30.0%, *p* < 0.001). A higher occurrence at night was observed in 2020 compared to 2014–2019 (16.6% vs. 11.7%, *p* = 0.001). Monthly and seasonal variabilities were observed in 2020. The months with the highest OHCA occurrence in 2020 were November (13.2%) and October (11.1%) and were significantly higher compared to the same months during the pre-pandemic period (9.1%, *p* = 0.002 and 7.9%, *p* = 0.009, respectively). Autumn was the season with the highest rate of OHCA, which was also higher compared to the pre-pandemic period (30.5% vs. 25.1%, *p* = 0.003). The COVID-19 pandemic was related to a higher occurrence of OHCA. The circadian, monthly, and seasonal variabilities of OHCA occurrence were confirmed. In 2020, the highest occurrence of OHCA was observed in October and November, which coincided with the highest occurrence of COVID-19 infections in Poland.

## 1. Introduction

The new coronavirus disease 2019 (COVID-19) pandemic, caused by severe acute respiratory syndrome coronavirus 2 (SARS-CoV-2), became a real challenge for humanity. The World Health Organization reported over 243 million confirmed COVID-19 cases and almost 5 million deaths caused by the disease worldwide as at 24 October 2021 [1]. The first confirmed case of COVID-19 in Poland was recorded on 4 March 2020, with over 3 million diagnosed infections since then [2]. COVID-19 infection, characterized by various stages, might lead to a severe inflammatory response caused by cytokine storm. This could result in acute respiratory insufficiency and multiorgan damage due to shock [3]. Furthermore, there is increasing evidence proving a relationship between COVID-19 and thromboembolic complications, which may result inter alia in myocardial infarction or sudden cardiac death [4].

Out-of-hospital cardiac arrest (OHCA) is a significant health issue worldwide. Despite improvements in treatment, it remains one of the major causes of death characterized by low survival rates [5]. Recently published studies showed lower survival rates in patients with OHCA during the pandemic era in comparison to the pre-pandemic era [6]. It has been proven that one of the basic methods to improve patient survival is to perform cardiopulmonary resuscitation as soon as possible [7]. This became more challenging during the pandemic era due to the risk of infection and the need for the healthcare workers to wear personal protective equipment [8]. The impact of COVID-19 on the incidence and outcome of OHCA was globally investigated in numerous studies; however, only a few compared the pandemic period with a pre-pandemic period that was longer than 1–2 years [6,9,10,11,12,13,14,15,16,17,18,19]. The majority of the studies along with metanalyses revealed an increased number of OHCA cases and a lower survival rate related to increased incidence of COVID-19 [6,15,17]. The Polish experience regarding OHCA and COVID-19 is rather scarce; furthermore, published reports focus predominantly on the early phase of the pandemic [20,21]. The occurrence of cardiac arrest is related to various factors including some environmental factors such as atmospheric conditions [22,23], air pollution [24,25], time variables [26,27,28]. Temporal variability, especially the circadian variation of OHCA occurrence, is a well-documented phenomenon and has been confirmed in a long-time observation period [27]. However, the temporal variability of OHCA occurrence within the COVID-19 pandemic in Poland remains uncertain.

The aim of the study was to investigate the chronobiology of OHCA during the COVID-19 pandemic and the differences in temporal variability of OHCA occurrence between the year 2020 and the pre-pandemic 6-year period.

## 2. Materials and Methods

The presented study was designed as a retrospective analysis of dispatch cards from the Emergency Medical Service in Opole, Poland, covering the period from January 2014 to December 2020. The dispatch cards were compatible with the Utstein characteristics. The presented research was a part of the OSCAR-POL registry whose methodology was described in detail in the previous publications [26,27]. Herein, we provide crucial features regarding the methodology. The registry covered an area of Opole district (1683 km^2^) with the population of approximately 261,000 inhabitants within the study period. The presumed cardiac etiology in patients over 18 years of age regardless of the initial rhythm, witnessed status or performing of CPR were the inclusion criteria. Late symptoms of death, known non-cardiac etiology, and OHCA in the pediatric population were the exclusion criteria.

OHCA occurrence was analyzed for the year 2020 and compared with the results from the 6-year pre-pandemic period covering the years 2014–2019. The temporal variation of OHCA occurrence was analyzed within the following patterns to stay consistent with previous reports. The circadian rhythm was investigated within 1 h periods as well as four 6 h intervals: “night” (00:00–05:59), “morning” (06:00–11:59), “afternoon” (12:00–17:59), and “evening” (18:00–23:59). Weekly, monthly, and seasonal variability were analyzed between consecutive days of the week, months of the year, and seasons, respectively. Four seasons were defined as spring (from 1 March till 31 May), summer (from 1 June till 31 August), autumn (from 1 September till 30 November), and winter (from 1 December till 28 or 29 February).

The study received the approval of the Ethics Committee of The Nicolaus Copernicus University in Torun, Collegium Medicum in Bydgoszcz (KB 471/2013) and was conducted in accordance with the Declaration of Helsinki and Good Clinical Practice principles.

A two-sided *p*-value < 0.05 was applied for statistical significance. The Shapiro–Wilk test and the analysis of histograms were performed to determine the data distribution. The differences between variables with non-normal distribution were analyzed using the Mann–Whitney test for two variables or the Kruskal–Wallis test for three or more variables. When the variables had normal distribution, the differences were tested with the Student t-test or ANOVA adequately to the number of the variables. Categorical variables were presented as absolute values and percentages. Despite the non-normal distribution of time variables in most cases it was decided to present the data as percentages and means with standard deviations (SD), and not medians with interquartile range to better visualize the differences. The statistical analysis was performed with the SPSS Statistic software version 28 (IBM Corp., Armonk, NY, USA).

## 3. Results

During the analyzed period, 3021 ambulance departures due to OHCA were observed of which 329 cases were excluded (16 cases occurred within the pediatric population and 313 referred to non-cardiac etiology, e.g., trauma or cancer). In the final analysis, 2692 cases were included. The mean age of all patients with OHCA within the study period was 70.2 ± 15.0 years with the majority male (63% vs. 37%, *p* < 0.001). The mean age of patients was 71.1 ± 14.7 in 2020 and 70.0 ± 15.0 (*p* = 0.221) in the years 2014–2019. Both during the pandemic and pre-pandemic periods almost two thirds of patients were male (63.3% vs. 63.0%, *p* = 0.903, respectively).

### 3.1. Temporal Variability of Out-of-Hospital Cardiac Arrest during 2020

The total number of OHCA cases reported in 2020 was 416. The circadian variability of the OHCA occurrence was observed with regard to division into both 1 h (Figure 1) and 6 h intervals (Figure 2) The highest number of OHCA cases was observed between 09:00 and 09:59 and the lowest between 03:00 and 03:59 (7.2% vs. 1.7%, *p* < 0.001, respectively). The histogram of the circadian distribution of OHCA occurrence presents a trimodal daily peak: between 09:00 and 10:59, between 13:00 and 14:59, and the last between 16:00 and 17:59. In the evening and during early night hours, the number of OHCA cases stayed at a relatively stable level with the nadir between 01:00 and 05:59. The night was associated with decreased number of OHCA cases (16.6%) in comparison to the afternoon (31.5%, *p* < 0.001) and the morning period (30.0%, *p* < 0.001). The difference between the total number of OHCA cases in the evening (21.9%) and in the night was insignificant (*p* = 0.127).

The distribution of OHCA occurrence in days of the week is presented in Figure 3. The lowest number of OHCA cases was observed on Monday (10.8%) with a gradually increasing number in the subsequent days reaching the highest value on Friday (16.8%). A slight decrease during the weekend was detected; however, in general the weekly variability did not reach statistical significance (*p* = 0.268).

Monthly (*p* = 0.017) and seasonal (*p* = 0.038) variability was observed during the year 2020. Figure 4 shows the monthly distribution of OHCA cases. The lowest OHCA occurrence was observed in September (6.3%; mean 0.87 ± 0.94) and was significantly lower in comparison to the highest observed in November (13.2%; mean 1.83 ± 1.15, *p* < 0.001) and October (11.1%; mean 1.48 ± 1.12, *p* = 0.029). The season with the highest proportion of OHCA cases (30.5%; mean 1.40 ± 1.13) was autumn, significantly higher compared to summer (21.2%; mean 0.96 ± 1.03, *p* = 0.031), the season with the lowest OHCA occurrence. The histogram of OHCA seasonal distribution is presented in Figure 5.

### 3.2. Comparison of Temporal Variability between the Year 2020 and 6-Year Pre-Pandemic Period (2014–2019)

The pre-pandemic period was characterized by a lower mean number of OHCA cases (1.04 ± 1.13 vs. 1.14 ± 1.07, *p* = 0.03). The median value of OHCA per year during that period was 379 with an interquartile range of 337 to 407. Circadian variability was confirmed for the year 2020 (*p* < 0.001) as well as for the 6-year pre-pandemic period (*p* = 0.018). Nevertheless, the distribution within a particular time of the day was different between those two time periods (Figure 2). In 2020, the highest percentage of OHCA cases occurred in the afternoon (31.5%); as opposed to the 6-year pre-pandemic period, where the highest percentage of OHCA cases occurred in the morning (32.7%). Higher occurrence during the night was observed in 2020 compared to the 2014–2019 period (16.6%; mean 1.64 ± 1.21 vs. 11.7%; mean 1.06 ± 1.12, *p* = 0.001) with no differences between other time periods during the day (Table 1). Weekly variability was not observed in the pre-pandemic period (*p* = 0.499), similarly to 2020 (Figure 3). OHCA occurred significantly more often on Sundays in 2020 in comparison to the pre-pandemic period. No differences were observed regarding the other days of the week (Table 1). Both analyzed periods were characterized by monthly (Figure 4) and seasonal (Figure 5) variability; however, the distribution of OHCA occurrence was different. In 2020, autumn was the season with the highest percentage of OHCA cases (30.5%) with a significantly higher occurrence of OHCA than in autumn during the pre-pandemic period (25.1%, *p* = 0.003). In the 2014–2019 period, winter was the season with the highest OHCA occurrence (27.2%), while in 2020 it was third, surpassing only summer in regard to the number of OHCA cases (Table 1). The months with the highest OHCA occurrence in 2020 were November (13.2%, mean 1.83 ± 1.15) and October (11.1%, mean 1.48 ± 1.12); the observed values were significantly higher in comparison to the same month during the pre-pandemic period (9.1%, mean 1.15 ± 1.13, *p* = 0.002 and 7.9%, mean 0.96 ± 0.99, *p* = 0.009, respectively). The highest OHCA occurrence in 2020 overlapped with the peak of the COVID-19 pandemic in Poland (Figure 6). No other differences regarding monthly OHCA occurrence were observed between the analyzed time periods (Table 1).

## 4. Discussion

The presented results showed the circadian, monthly, and seasonal rhythm of OHCA occurrence with no weekly variability in the year 2020 or during the 6-year pre-pandemic period (2014–2019). Nevertheless, the pattern of OHCA occurrence was different in the compared time periods as the mean value of OHCA occurrence was higher during the pandemic year. In 2020, OHCA was observed significantly more often at night, on Sundays, and in autumn, particularly in October and November.

Previous studies delivered evidence of the association between a higher risk of OHCA occurrence and the epidemics of particular infectious diseases such as influenza [29]. The global spread of the SARS-CoV-2 virus is an exceptional challenge for medical staff; hence, its relationship with OHCA occurrence is of great importance. In our study, we observed a higher mean number of OHCA cases during the year 2020 in comparison to the pre-pandemic period. This observation is in line with the majority of previous reports from the United Kingdom [12,30], Lombardy, Italy [10], Singapore [14], the United States [13,17,31,32], and Paris, France [16]. In contrast, studies from Australia [18], Switzerland [33], and Padua, Italy [34] showed no differences in OHCA incidence between pre-pandemic and pandemic eras. Data from Western Australia showed a similar incidence of OHCA cases during the period with the most stringent restrictions due to the COVID-19 pandemic compared to the previous year [18]. However, as underlined by the authors, during that period, a relatively low incidence of COVID-19 was reported (107 cases per 100,000 person-years). The findings indirectly support the observations from other regions with high COVID-19 incidence, where the OHCA occurrence was higher in comparison to the pre-pandemic period. Interestingly, a population-based, observational study from Switzerland showed an increased incidence of OHCA cases in 2020 in Cantons with low COVID-19 morbidity and an opposite situation in Cantons with high COVID-19 incidence [33]. The meta-analysis by Lim et al. [15] included 35,379 OHCA cases from 10 studies, showed a 120% increase in cardiac arrest occurrence regardless of etiology. However, the difference in total OHCA occurrence was predominantly related to a significant increase in arrests caused by trauma (OR 1.69, 95% CI 1.07–2.69, *p* = 0.03), which were excluded from our study. To date, the majority of published studies focused on the OHCA occurrence during the first wave of the COVID19 outbreak and little is known about later developments of the pandemic. Baldi et al. [35] analyzed all OHCA cases in the provinces of Pavia, Lodi, Cremona, Mantua, and Varese in Italy between February and December 2020 and divided this period into two, reflecting the first and second wave of the pandemic. The authors observed a significant positive correlation between increased COVID-19 incidence and the incidence of OHCA cases. Nevertheless, it should be noted, that the correlation was observed only in the provinces most affected by the pandemic and no relationship for provinces with low COVID-19 incidence was found.

In the meta-analysis by Borkowska et al. [6] that included 4210 patients with OHCA, the authors reported lower survival to hospital discharge in the COVID-19 group (OR 0.25, 95% CI 0.12–0.53, *p* < 0.001) along with a lower percentage of shockable rhythms (5.7% vs. 37.4%). The same observation was confirmed in the previously mentioned meta-analysis by Lim et al., which presented a 63% higher chance of achieving the return of spontaneous circulation and a 65% higher chance of survival to hospital discharge before the pandemic [15]. However, some single studies did not show a lower chance of survival during the pandemic. The observational study based on the OHCA registry in Daegu, South Korea, revealed a similar number of OHCA cases of cardiac etiology in the pandemic and pre-pandemic periods (n = 152 vs. n = 142) with no difference in survival to hospital discharge, yet, with a significantly lower rate of patients with a good neurologic outcome (OR 0.23, 95% CI 0.05–0.98) [9]. Results based on a registry from Taiwan also showed a similar rate of survival to discharge between February and April 2020 in comparison to the same period in 2019 (4.98% vs. 5.96%); however, with a significantly lower rate of favorable neurologic outcome (2.09% vs. 4.21%, *p* = 0.035). Potential causes for lower survival could be related to a lower percentage of shockable rhythms, bystander CPR, the longer response time of the emergency medical services, or higher rates of OHCA at home [15,16,36]. The need to wear full personal protective equipment by medical personnel might also be a significant factor for unfavorable outcomes [8].

Polish data regarding the relationship between COVID-19 incidence and OHCA occurrence is scarce. Borkowska et al. [20] performed an observational study of OHCA cases in Masovian Voivodeship between March and April 2020. A total of 527 OHCA cases were analyzed, of which 379 had cardiac etiology. The authors reported OHCA incidence of 0.12/1000 inhabitants during the study period. This result is lower than the incidence reported in previous studies from Poland where it was estimated to be 0.57–1.70 per 1000 inhabitants per year [26,37,38]. It should be noted that the authors analyzed OHCA occurrence within the first two months of the pandemic in Poland, when the total number of COVID-19 infections was very low and strict lockdown restrictions were present. Results from the Polish Registry of Acute Coronary Syndromes showed a decreased number of patients admitted to hospital due to acute coronary syndrome with and without ST-segment elevation during the first wave of the COVID-19 pandemic (March–May 2020) compared to the same period in 2019 [21]. At the same time, the rate of OHCA was higher in 2020 (3.43% vs. 2.75%, *p* = 0.049) in the population of patients with myocardial infarction. The COVID-19 pandemic also affected patients with heart failure. Kubica et al. [39] showed a reduction in hospital admissions due to acute heart failure by over 23% in the year 2020 with an increase in in-hospital all-cause mortality (6.5% vs. 5.2%, *p* < 0.001).

To the best of our knowledge, this is the first study to focus on the temporal variability of OHCA during the first year of the COVID-19 pandemic. Similar to the results from previous studies, the circadian, monthly, and seasonal variabilities of OHCA occurrence were confirmed in 2020; however, a different pattern was observed [27]. The OHCA rate was significantly higher at night in 2020 compared to the pre-pandemic period. The explanation of this phenomenon requires further study. A potential explanation might be related to the increased number of emergency calls in 2020, especially during the night shift (an increase of almost 80%), which might reflect the general alertness of the population due to new medical threats [40]. In autumn 2020, especially during October and November, the very peak of COVID-19 infections was observed in Poland [41]. This period overlaps with the highest rates of OHCA cases, which were significantly higher compared to the pre-pandemic period. The observation stays in line with the previous studies showing a correlation between increased OHCA cases and increased SARS-CoV-2 infection rates [17,35].

The main limitation of the study is the retrospective design and the lack of detailed data regarding, e.g., patients’ medical history or socio-economic status. Secondly, no analysis of witnessed status, initial rhythm, or survival was performed. In the presented study, the year 2020 was considered as the pandemic period, although the first COVID-19 case diagnosed in Poland was diagnosed later than January 2020. A similar approach was also used in previous studies [13]. Therefore, the analysis of seasonal variability of OHCA occurrence in the light of SARS-CoV-2 infections, particularly in the winter, is limited. Still, the main objective of the study was to assess the chronobiology of OHCA occurrence in the year 2020 affected by the COVID-19 pandemic and not to investigate potential influencing factors. The study showed OHCA occurrence in 2020 and the 6-year pre-pandemic period; however, the results reflected the situation on a relatively small territory of Opole district. Due to a relatively low number of cases presented, especially in the year 2020, observed differences should be confirmed on a larger group. Therefore, further national-based studies should be conducted in Poland.

## 5. Conclusions

The COVID-19 pandemic was related to a higher occurrence of OHCA in comparison to the 6-year pre-pandemic period. The circadian, monthly, and seasonal variabilities of OHCA occurrence were confirmed, both in the year 2020 and in the 2014–2019 period with no differences between the weekdays. A higher number of OHCA cases occurred at night and on Sundays during the pandemic year. In 2020, the highest occurrence of OHCA was observed in autumn, especially in October and November, which coincided with the highest occurrence of COVID-19 infections in Poland during the second wave of the pandemic. Further studies should be performed to explore the long-term impact of SARS-CoV-2 infection on the incidence of OHCA.

## Figures and Tables

**Figure 1 jcm-11-04143-f001:**
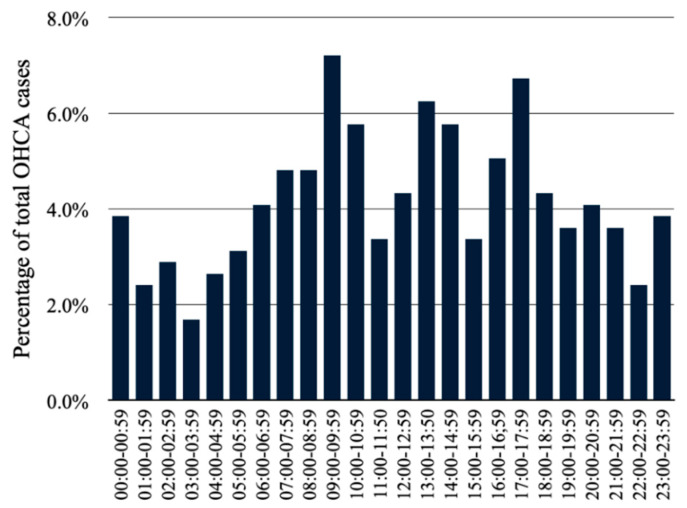
Circadian distribution of out-of-hospital cardiac arrest (OHCA) occurrence divided into 1 h intervals in 2020 (*p* < 0.001).

**Figure 2 jcm-11-04143-f002:**
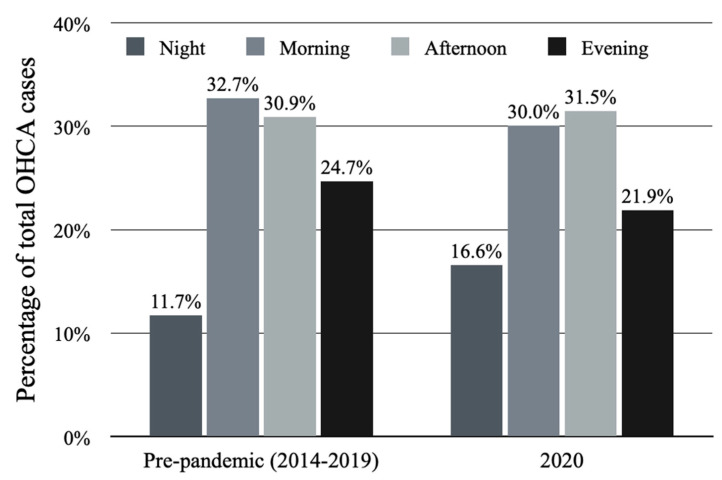
Circadian distribution of out-of-hospital cardiac arrest (OHCA) occurrence in 6 h intervals within 24 h in year 2020 (*p* < 0.001) and during 2014–2019 pre-pandemic period (*p* < 0.001).

**Figure 3 jcm-11-04143-f003:**
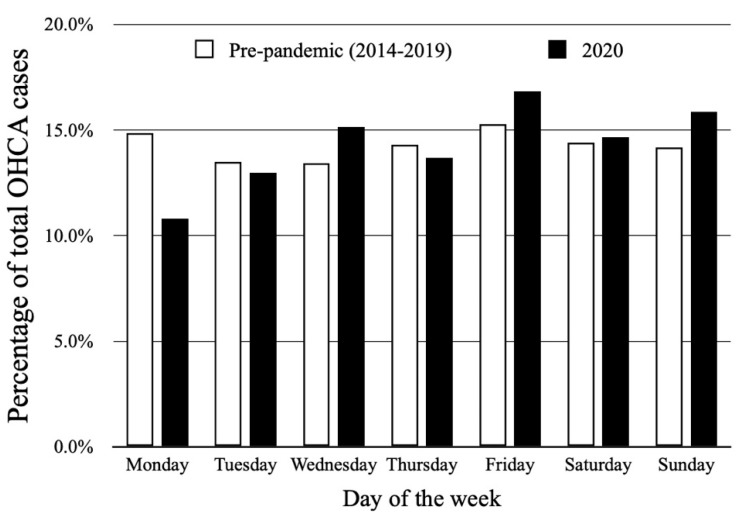
Distribution of out-of-hospital cardiac arrest (OHCA) occurrence in subsequent days of the week in year 2020 (*p* = 0.268) and during 2014–2019 pre-pandemic period (*p* = 0.499).

**Figure 4 jcm-11-04143-f004:**
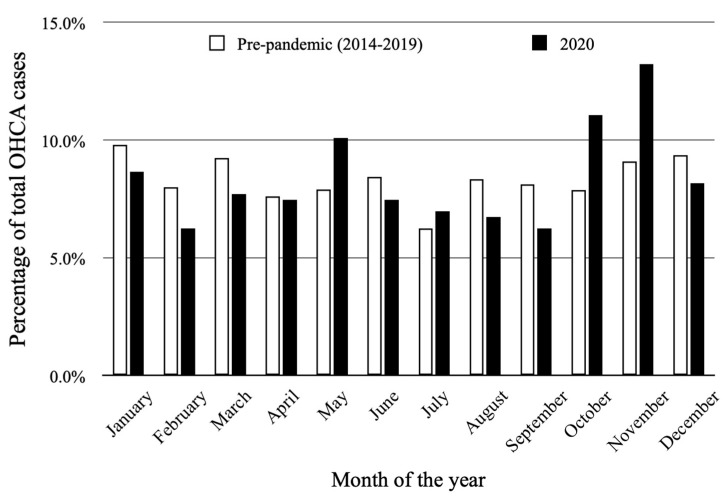
Monthly distribution of out-of-hospital cardiac arrest (OHCA) occurrence in year 2020 (*p* = 0.017) and during 2014–2019 pre-pandemic period (*p* = 0.009).

**Figure 5 jcm-11-04143-f005:**
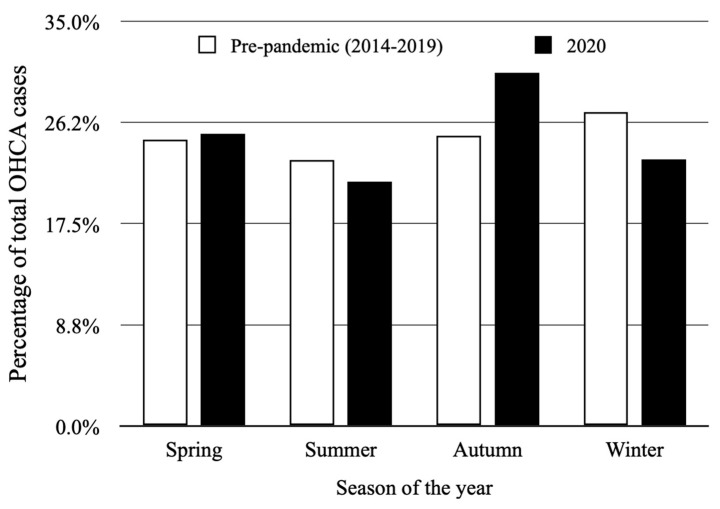
Seasonal distribution of out-of-hospital cardiac arrest (OHCA) occurrence in year 2020 (*p* = 0.038) and during 2014–2019 pre-pandemic period (*p* = 0.021).

**Figure 6 jcm-11-04143-f006:**
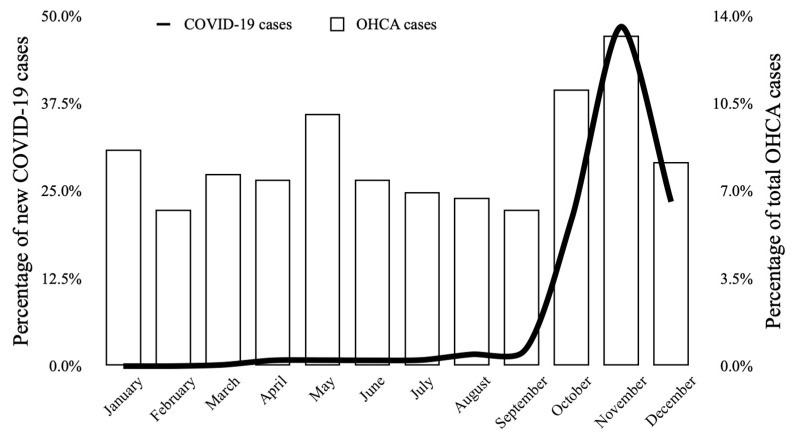
Monthly distribution of out-of-hospital cardiac arrest (OHCA) occurrence and new COVID-19 cases diagnosed in Poland in year 2020. Black line represents the percentage of new COVID-19 cases diagnosed in a particular month (data based on the daily reports of The Ministry of Health of Poland).

**Table 1 jcm-11-04143-t001:** Comparison of differences in temporal variability of OHCA occurrence between pre-pandemic period and year 2020.

Temporal Variables	2020 Mean ± SD	Pre-Pandemic Period (2014–2019) Mean ± SD	*p*-Value
Time of the day			
Night	1.06 ± 1.12	1.64 ± 1.21	0.001
Morning	2.96 ± 1.85	2.98 ± 1.70	0.80
Afternoon	2.79 ± 1.67	3.12 ± 1.81	0.35
Evening	2.23 ± 1.68	2.17 ± 1.59	0.96
Day of the week			
Monday	0.87 ± 0.95	1.08 ± 1.12	0.24
Tuesday	1.04 ± 0.99	0.98 ± 1.10	0.47
Wednesday	1.19 ± 1.18	0.98 ± 1.11	0.17
Thursday	1.08 ± 1.11	1.04 ± 1.08	0.85
Friday	1.35 ± 1.12	1.11 ± 1.11	0.12
Saturday	1.17 ± 1.17	1.05 ± 1.22	0.36
Sunday	1.27 ± 0.93	1.03 ± 1.15	0.026
Month of the year			
January	1.16 ± 1.16	1.20 ± 1.21	0.95
February	0.90 ± 0.82	1.08 ± 1.09	0.58
March	1.03 ± 1.11	1.13 ± 1.22	0.81
April	1.03 ± 0.93	0.96 ± 1.03	0.51
May	1.35 ± 1.14	0.97 ± 0.99	0.06
June	1.03 ± 1.03	1.07 ± 1.13	0.99
July	0.94 ± 1.03	0.76 ± 1.06	0.29
August	0.90 ± 1.04	1.02 ± 1.16	0.66
September	0.87 ± 0.94	1.03 ± 1.18	0.65
October	1.48 ± 1.12	0.96 ± 0.99	0.009
November	1.83 ± 1.15	1.15 ± 1.13	0.002
December	1.10 ± 1.04	1.15 ± 1.27	0.89
Season of the year			
Spring	1.14 ± 1.07	1.02 ± 1.08	0.22
Summer	0.96 ± 1.03	0.95 ± 1.12	0.73
Autumn	1.40 ± 1.13	1.05 ± 1.1	0.003
Winter	1.05 ± 1.02	1.14 ± 1.19	0.77

## Data Availability

Data are available upon reasonable request. All data relevant to the study are included in the article. The original data are available from the corresponding author, within the limits of the signed informed consent from the contributors.

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
