# Peer review of "Occurrence and Temporal Variability of Out-of-Hospital Cardiac Arrest during COVID-19 Pandemic in Comparison to the Pre-Pandemic Period in Poland—Observational Analysis of OSCAR-POL Registry"

_jcm, 2022, doi:10.3390/jcm11144143_

Round 1

Reviewer 1 Report

I have been honored to review this interesting article entitled” Occurrence and temporal variability of out-of-hospital cardiac 2 arrest during COVID-19 pandemic in comparison to the pre-3 pandemic period in Poland”.

Authors studied the eventual relationship between COVID-19 and out-of-hospital cardiac arrest (OHCA) through a chronobiologic analysis of OHCA occurrence during the pandemic era in comparison to the pre-pandemic era.

Major concern

-          The pandemic era covered a 9-month period (since the first confirmed case of COVID-19 was reported on March 4th 2020). Do the authors have data over the whole year including the first months of the 2021?

-          The 2020 winter should be included in the pre-pandemic era since non COVID cases were included in the study.

-          According to the authors, “In 2020 autumn was the season with the highest percentage of OHCA cases (30.5%) with significantly higher occurrence of OHCA than in autumn during the pre-pandemic period (25.1%, p=0.003)” but as above mentioned, winters (that is supposed to be the most dangerous season for OHCA occurrence) were not comparable.

-          Data on OHCA in COVID and not affected patients could make the manuscript more interesting to evaluate the potential role of thromboembolic complications observed worldwide.

-          Data on intra and extra-hospital mortality in OHCA patients could be interesting to evaluate the different deployment of medical resources.

Minor concern

-          How many hospitals/clinics have an ED in the Opole district?

-          Figure 1a is not very clear, could the authors modify the picture to make the circadian OHCA distribution clearer?

Author Response

Response to Reviewer 1 Comments:

“I have been honored to review this interesting article entitled” Occurrence and temporal variability of out-of-hospital cardiac 2 arrest during COVID-19 pandemic in comparison to the pre-3 pandemic period in Poland”.

Authors studied the eventual relationship between COVID-19 and out-of-hospital cardiac arrest (OHCA) through a chronobiologic analysis of OHCA occurrence during the pandemic era in comparison to the pre-pandemic era.

Point 1: The pandemic era covered a 9-month period (since the first confirmed case of COVID-19 was reported on March 4th 2020). Do the authors have data over the whole year including the first months of the 2021?

Point 2: The 2020 winter should be included in the pre-pandemic era since non COVID cases were included in the study.

Point 3: According to the authors, “In 2020 autumn was the season with the highest percentage of OHCA cases (30.5%) with significantly higher occurrence of OHCA than in autumn during the pre-pandemic period (25.1%, p=0.003)” but as above mentioned, winters (that is supposed to be the most dangerous season for OHCA occurrence) were not comparable.

Response 1, 2, and 3:

Thank you for these thoughtful remarks. We decided to answer to those remarks in one point because all the points refer to the same issues mentioned by the Reviewer 1. Unfortunately, we currently do not have the data covering the first months of the year 2021 [Response 1].

We decided to define the 2020 as pandemic period as the similar approach was applied in previous studies regarding the relationship between OHCA and the pandemic. Glober et al. (Am J Emerg Med. 2021 Oct; 48: 191–197. doi: 10.1016/j.ajem.2021.04.072) presented the data from the US. The authors defined the period from January 1, 2020 to June 30, 2020 as the pandemic period even though the first COVID-19 case was diagnosed in the US at the end of January [Response 2].

According to our data the autumn was the season with the highest OHCA occurrence during the pandemic period according to the definition used in the study. This approach to data analysis can be seen as a limitation of our research, however. as mentioned above we do not have data including first months of 2021. Therefore, we cannot meet the reviewer's expectations to adopt a different time frame for our study. We decided to add the information regarding the comparability of the winter season and the lack of data covering 2021 in the limitations of the manuscript.

Point 4: Data on OHCA in COVID and not affected patients could make the manuscript more interesting to evaluate the potential role of thromboembolic complications observed worldwide.

Response 4: Thank you very much for this remark. We strongly agree that the comparison of COVID-19-positive and COVID-19-negative patients would be very interesting, however that was not the objective of the current analysis. The aim of our study was to investigate the chronobiology of the OHCA occurrence especially during the pandemic period and the potential differences in comparison to the previous years. Furthermore, patients with OHCA were not routinely diagnosed for COVID-19 infection, especially during the first months of the pandemic due to lack of the fast test available for the Emergency Medical Teams to perform on the scene. For this reason, the analysis suggested by the Reviewer, although very interesting, would be impossible to perform.

Point 5: Data on intra and extra-hospital mortality in OHCA patients could be interesting to evaluate the different deployment of medical resources.

Response 5: We agree that the analysis of intra and extra-hospital mortality would be interesting. However, as mentioned in previous responses, that was not the aim of our study. Our data come from a central Emergency Medical Service station in Opole that is responsible for departures of all Emergency Medical Teams in Opole district and unfortunately, we do not have the data regarding follow-up of the patients who survived to the hospital admission.

Point 6: How many hospitals/clinics have an ED in the Opole district?

Response 6: There are 8 emergency departments in the Opole district. However, there is one central Emergency Medical Service station with 10 Emergency Medical Teams that serves for all above mentioned EDs.

Point 7: Figure 1a is not very clear, could the authors modify the picture to make the circadian OHCA distribution clearer?

Response 7: Thank you for this remark. We have separated Figure 1a and 1b and created two separate Figures (Figure 1 and Figure 2) to increase the visibility of the data.

Reviewer 2 Report

In their paper „Occurrence and temporal variability of out-of-hospital cardiac arrest during COVID-19 pandemic in comparison to the pre-pandemic period in Poland – observational analysis of OSCAR-POL registry”, Jakub Ratajczak describe the occurrence of out-of-hospital cardiac arrest (OHCA) due to a cardiac cause over the timespan 2014-2020 in a Polish district with approximately 261,000 inhabitants. OHCA due to other causes (traumatic, cancer) were excluded. The compared numbers of OHCA at different times of the day, seasons of the year and, most importantly, between the COVID era (2020) and the previous years (2014-2019). They found generally higher numbers of OHCA in 2020, and the numbers match the number of COVID cases with a drastic peak in autumn, that exceeds previous years by far. They conclude that seasonal changes of OHCA incidence exit and that COVID was associated with an increased occurrence of OHCA.

This analysis is severely limited by the fact that the authors only describe the incidence of OHCA and no other clinical variables. It is unclear to me why the authors did not analyse clinical parameters that should be available as the authors used “Utstein style-like” documentation forms. Furthermore, clinical follow-up and the presence of a COVID infection would have been very helpful, but this data may be difficult to get.

Otherwise, this is a well-written manuscript with well-performed statistics and the conclusions are congruent to the results. References are adequate.

Minor comments:

-        Abstract: The exact increase of OHCA during COVID should be mentioned in the abstract, if possible.

-        Line 108: “This” should be removed at the begin of the Results.

-        Line 151: What does “1.40 +/- 1.13” refer to?

-        The authors write that the incidence of acute coronary syndrome (ACS) decreased during COVID. Could the increased rate of OHCA also partly be explained by patients with ACS not presenting to the hospital?

Author Response

Response to Reviewer 2 Comments:

“In their paper „Occurrence and temporal variability of out-of-hospital cardiac arrest during COVID-19 pandemic in comparison to the pre-pandemic period in Poland – observational analysis of OSCAR-POL registry”, Jakub Ratajczak describe the occurrence of out-of-hospital cardiac arrest (OHCA) due to a cardiac cause over the timespan 2014-2020 in a Polish district with approximately 261,000 inhabitants. OHCA due to other causes (traumatic, cancer) were excluded. The compared numbers of OHCA at different times of the day, seasons of the year and, most importantly, between the COVID era (2020) and the previous years (2014-2019). They found generally higher numbers of OHCA in 2020, and the numbers match the number of COVID cases with a drastic peak in autumn, that exceeds previous years by far. They conclude that seasonal changes of OHCA incidence exit and that COVID was associated with an increased occurrence of OHCA.”

Point 1: This analysis is severely limited by the fact that the authors only describe the incidence of OHCA and no other clinical variables. It is unclear to me why the authors did not analyse clinical parameters that should be available as the authors used “Utstein style-like” documentation forms. Furthermore, clinical follow-up and the presence of a COVID infection would have been very helpful, but this data may be difficult to get.

Otherwise, this is a well-written manuscript with well-performed statistics and the conclusions are congruent to the results. References are adequate.

Response 1: Thank you very much for your thoughtful analysis of our manuscript and for this remark. We decided to present and focus on data regarding occurrence of OHCA due to two main reasons. Firstly, and most importantly, from the beginning the study was aimed at presenting the phenomenon of OHCA chronobiology and temporal variability and not potential influencing factors. We strongly agree that the clinical parameters, follow-up, and the information about COVID infection would be very interesting, however it was not the purpose of our study. We have underlined that in the limitations of the study at the end of the discussion. Secondly, as also mentioned in the limitations, the retrospective character of the study limited the availability of various parameters.

Point 2: Abstract: The exact increase of OHCA during COVID should be mentioned in the abstract, if possible.

Response 2: Thank you for this suggestion. We have added the information regarding the median and the interquartile range of the OHCA cases during the pre-pandemic period to the abstract and the main text of the manuscript. The direct, statistical comparison in this case is problematic because we have only one absolute value of OHCA cases in 2020 and multiple values during the pre-pandemic period.  

Point 3: Line 108: “This” should be removed at the begin of the Results.

Response 3: As suggested by the Reviewer the demonstrative “This” has been removed.

Point 4: Line 151: What does “1.40 +/- 1.13” refer to?

Response 4:  In the sentence “The season with the highest proportion of OHCA cases (30.5%; mean 1.40 ± 1.13) was autumn, (…)” the value “1.40 +/- 1.13” refers to the mean daily number of OHCA cases within a particular season, in this case autumn.

Point 5: The authors write that the incidence of acute coronary syndrome (ACS) decreased during COVID. Could the increased rate of OHCA also partly be explained by patients with ACS not presenting to the hospital?

Response 5: Thank you for this thoughtful remark. We agree that the increased rate of OHCA cases of cardiac etiology could be related and partly explained by decreased number of ACS patients treated in the hospital. Due to many pandemic-related reasons (fear of infection, structural changes within the medical units) patients delayed their medical appointments or undervalue significant symptoms.

Round 2

Reviewer 1 Report

I really appreciated the effort to improve the manuscript's quality.

Lack of data covering the first months of the year 2021 are a limitation but threre is no solution according to the authors reply.